# Preservation of *Mimosa tenuiflora* Antiaflatoxigenic Activity Using Microencapsulation by Spray-Drying

**DOI:** 10.3390/molecules27020496

**Published:** 2022-01-13

**Authors:** Christopher Hernandez, Laura Cadenillas, Céline Mathieu, Jean-Denis Bailly, Vanessa Durrieu

**Affiliations:** 1Toxalim (Research Center in Food Toxicology), Université de Toulouse, INRAE, ENVT, EI-Purpan, 31300 Toulouse, France; hernandezhernandezchristopher@gmail.com (C.H.); laura.cadenillas.s@gmail.com (L.C.); 2Laboratoire de Chimie Agro-Industrielle (LCA), Université de Toulouse, INRA, INPT, 4 allée Emile Monso, 31030 Toulouse, France; vanessa.durrieu@ensiacet.fr; 3Centre d’Application et de Traitement des Agro-Ressources (CATAR), INPT, 4 allée Emile Monso, 31030 Toulouse, France; celine_mathieu@hotmail.com; 4Ecole Nationale Vétérinaire de Toulouse, 23 Chemin des Capelles, CEDEX, 31076 Toulouse, France

**Keywords:** encapsulation, spray-drying, antiaflatoxin activity, antioxidant activity, polysaccharides

## Abstract

*Mimosa tenuiflora* aqueous extract (MAE) is rich in phenolic compounds. Among them, condensed tannins have been demonstrated to exhibit a strong antioxidant and antiaflatoxin B1 activities in *Aspergillus flavus*. Since antioxidant capacity can change with time due to environmental interactions, this study aimed to evaluate the ability of encapsulation by spray-drying of *Mimosa tenuiflora* aqueous extract to preserve their biological activities through storage. A dry formulation may also facilitate transportation and uses. For that, three different wall materials were used and compared for their efficiency. Total phenolic content, antioxidant activity, antifungal and antiaflatoxin activities were measured after the production of the microparticles and after one year of storage at room temperature. These results confirmed that encapsulation by spray-drying using polysaccharide wall materials is able to preserve antiaflatoxin activity of *Mimosa tenuiflora* extract better than freezing.

## 1. Introduction

Aflatoxin B1 (AFB1) is a worldwide public health issue due to its carcinogenic, immunotoxic and mutagenic properties in humans and animals [1,2]. This toxic fungal metabolite is produced by several species of *Aspergillus*, mostly grouped in the *Flavi* section [3]. Aflatoxins are thermophilic species and AFB1 is a major contaminant in regions with hot climate. It is estimated that almost 4.5 billion people are exposed regularly to this contaminant through their diet, leading to an important number of DALYs (death and disability adjusted life years) [4]. Moreover, global warming is increasing the geographic repartition of AFB1 that was recorded in regions usually considered as free, such as Europe [5]. Aflatoxigenic species can grow and produce toxins either in the field or later during storage, and many crops can be contaminated [6,7]. Due to the toxicity and especially to the carcinogenic property of AFB1, many countries have set up regulation in different foods to limit consumers’ exposure [8].

Many works are devoted to identifying strategies to limit AFB1 contamination, detoxifying the toxin when present or limiting its deleterious effects after ingestion [7].

Due to the great stability of the molecule, the best way to protect consumers from its toxicity seems to be avoiding its presence. Among developing strategies aiming to reduce AFB1 contamination of crops, the use of atoxigenic strains to compete with toxigenic ones was deeply studied [9]. However, this strategy, that leads to interesting results in the short term, could be questionable in the long term due to the risk of recombination between atoxigenic and toxigenic strains [10]. It also raises the question of its impact on biodiversity.

The use of plant extracts to limit aflatoxigenic species development or AFB1 production is another strategy, and has been increasingly studied over the past 5 years. Several plant extracts, essential oils or isolated components were found able to block aflatoxin B1 production [11,12], among them *Mimosa tenuiflora* aqueous extract (MAE). Contrary to what is observed using essential oils that inhibit AFB1 synthesis together with reducing fungal growth [13], MAE was shown to inhibit AFB1 in *A. flavus* with no impact on fungal development [14]. This result is interesting because if the extract is able to block AFB1 production without modifying fungal growth, its impact on biodiversity shall be limited. It also reduces the risk of resistance development in target species, since it does not create a selective pressure [15]. The effect of MAE was linked to the presence of condensed tannins, in agreement with results observed on mycotoxin production using some wood/forest and vine by-products [16].

In many cases, a positive correlation could be demonstrated between antioxidative ability of plant extracts/compounds and their impact on AFB1 synthesis [11,14,17,18]. This observation raises the question of the preservation of their efficacy with time. Indeed, antioxidative potential can change with time and be reduced due to environmental conditions during storage. Different important factors have been identified, among which are light, temperature, pH, and oxygen [19,20].

Therefore, the aim of this study was to evaluate the impact of microencapsulation using different wall materials on the antioxidative and antiaflatoxigenic ability of MAE.

Indeed, microencapsulation was recently proposed as a final processing step that can be used to stabilize and protect antioxidant compounds during storage and application [21,22]. Through encapsulation, the active ingredients are trapped within the wall materials, limiting their interaction with environmental factors [21,23].

This process can also allow for the development of a dry formulation of an active extract. Regarding the limitation of mycotoxin contamination during storage, such dry formulation would be more compatible with its use on dried grains, the active compounds only being released in case of moistening of the medium and subsequent increased risk of mycotoxin production.

Encapsulation depends on several factors, mainly the type of core and wall materials and the technique used. The most widely used encapsulation techniques are spray-drying (SD) and freeze-drying (FD). Different types of biopolymers can be used as wall materials, such as maltodextrin (MD), starch, Arabic gum, and proteins [24].

Within this context, we encapsulated MAE using the spray-drying method and three different wall materials: maltodextrin, starch, and soy protein. We evaluated the impact of encapsulation process on the antioxidant, antifungal and antiaflatoxigenic effects of MAE. To evaluate the ability of such methods to maintain these activities with time, they were again evaluated after one year of storage at room temperature.

## 2. Results and Discussion

### 2.1. Biological Activities of the MAE and Evolution after One-Year Storage (sMAE)

The characterization of the aqueous extract of *Mimosa tenuiflora* bark at time 0 and after one year of storage at −20 °C (sMAE) is shown in Table 1.

After one-year storage, no significant changes in the total amount of phenolic compounds nor the antioxidant activity were observed in sMAE. This may be due to the low temperature of storage (−20 °C), which helps to preserve the stability of the phenolic compounds. However, regarding AFB1 inhibition, the ability of sMAE to inhibit AFB1 production was strongly reduced. Indeed, maximum AFB1 inhibition went from 65.7% at time 0 to 43.8% after one year. This result highlights the need to improve the storage procedure to maintain biological activity of MAE.

### 2.2. MAE Spray-Drying Encapsulation

The process yields obtained during spray-drying step both for microparticles with and without MAE, are shown in Table 2.

Soy protein isolate showed the lowest spray-drying yield at 57%, followed by starch (58%) and maltodextrin (59%). The spray-drying yields obtained for the microparticles containing the MAE followed the same trend. As for microparticles without MAE, the highest yield was obtained for maltodextrin-based microparticles (63%), followed by starch-based ones (59%) and soy protein-based ones (54%). Spray-drying yield is not frequently reported for natural compounds. It appears that our results are slightly higher than the few previously reported, such as the 51% described for the encapsulation of aqueous bitter melon extract by spray-drying with maltodextrin [25], or the 49% in the case of microparticles prepared with OSA starch to encapsulate Jussara pulp [26].

These yields can be considered promising regarding the use of this process at an industrial scale for the protection of bioactive compounds, especially considering the low amount of dry matter used during the atomization step.

### 2.3. Particles Characterization

#### 2.3.1. Particles Morphology and Size

The macroscopic aspect of the produced microparticles is shown in Figure 1.

The microparticles produced using maltodextrin were the most homogeneous and appeared lighter in color than the others (Figure 1A). When MAE was incorporated, a brown color was observed for all wall materials (Figure 1D–F). As noted for microparticles made only with support material, those made with maltodextrin appeared mildly lighter and thinner than the others (Figure 1D).

A morphological analysis was carried out to evaluate the characteristics of the microparticles such as shape, surface, as well as their size distribution (Figure 2). Scanning electron microscopy showed that the microparticles are globally spherical, heterogeneous in size and not agglomerated, which is probably related to repulsion by negative charges.

Figure 2 shows that the microparticles had a wrinkled surface, probably because they collapsed in their center. This observation can be explained by the evaporation of water contained in droplets inside the drying chamber during the spraying step [27]. This phenomenon was previously observed in plant protein-based microparticles with low dry matter content [28]. This could also be related to SEM analysis, which is performed under vacuum which can deform the microparticles [29].

The sizes of microparticles made using polysaccharides as wall material (maltodextrin and starch) were about 3 µm in size. Those based on the use of soy protein were larger and had a less homogeneous size distribution, ranging between 10 and 15 µm and reaching up to 30 μm in some cases. There could be a link between the intrinsic size of the matrix constituent macromolecules and the size of the resulting microparticles. Furthermore, soy protein was more difficult to dissolve than the others; the large particle visible on Figure 2C could be the result of this incomplete dissolution.

#### 2.3.2. Total Polyphenol Content and Antioxidant Activity of Wall Material Microparticles

The total polyphenol content and antioxidant activity of the particles made with wall materials only are presented on Table 3.

None of the microparticles with support material presented antioxidant activity. Total polyphenol content in microparticles made with the two polysaccharides (maltodextrin and starch) was not significant. By contrast, the soy protein microparticles showed a polyphenol content of 17 mg GAE/g DM. This value is attributed to the fact that the Folin reagent, used for polyphenol determination, can also react with proteins, as used in the Lowry method for protein determination [30]. However, this value was low compared to values measured in MAE. Moreover, if due to protein interactions, this value shall remain constant and not interfere with the evaluation of the evolution of polyphenol content in microparticles containing MAE with storage time.

#### 2.3.3. Total Polyphenol Content and Antioxidant Capacity of Microparticles with MAE and Evolution after 1 Year of Storage

Table 4 gathers the results of the total polyphenols and antioxidant capacity of the microparticles containing MAE on time 0 and after one year of storage at room temperature. The determination of the polyphenols encapsulation efficiency (PEE) allowed us to demonstrate the capacity of the tested wall materials to preserve the polyphenols during the atomization stage.

For all the microparticles, the experimental results of the polyphenol determination were lower than the theoretical ones. This could be explained by the matter loss on the apparatus walls during the encapsulation process (both some wall material and compounds belonging to the extract).

Polysaccharide-based microparticles had the highest polyphenols encapsulation efficiency, with 86% and 65% for starch and maltodextrin, respectively. Soy protein-based ones had the lowest efficiency, with 58%. This difference may be due to the polysaccharides’ hydrophilic character, making their affinity for an aqueous extract higher than that of soy protein, which is more hydrophobic. The difference between the two polysaccharides may be related to the longer chain molecular structure of starch that can improve the phenolic compounds protection. The microparticles’ storage for one year at room temperature did not significantly modify the polyphenol content, whatever the wall material considered.

Only few data are available concerning the encapsulation of natural extracts. Nevertheless, we can compare our results with those obtained during the microencapsulation of phenolic compounds of pomegranate juice by spray-drying, where the microparticles generated with maltodextrin presented a lower polyphenol encapsulation efficiency (54%). By contrast, the microparticles generated with soy protein presented a higher efficiency (76%) [31]. Our results follow the same trend as those reported for the encapsulation of polyphenols of horseradish juice by spray-drying with maltodextrin (75%), starch (77%) and soy protein (56%) [22]. Our results are also comparable with the results described for anthocyanins’ encapsulation by spray-drying with sodium alginate as wall material (75%) [21] and of carotenoids by spray-drying with a mixture of maltodextrin and Arabic gum, which presented an efficiency of 77% [32].

The antioxidant capacity of MAE was not modified by encapsulation with the two polysaccharide-based microparticles. Indeed, considering the 50/50 ratio (*w/w*) between the extract and the wall material used for encapsulation, which led to a division of the DM of MAE by two-fold in the microparticles, it seems that the initial antioxidant activity of the aqueous extract was preserved. Moreover, it was not significantly affected by one-year storage at room temperature (Table 4). By contrast, the antioxidant activity of MAE in soy protein-based microparticles was twice lower than that for polysaccharides. This may be due to the fact that antioxidant compounds can interact with proteins, reducing their activity. This observation is even more important after one year of storage, since antioxidant activity decreased again about two-fold, increasing the difference to polysaccharide-based microparticles.

### 2.4. Antifungal and Antiaflatoxin Activities

#### 2.4.1. Antifungal and Antiaflatoxin Activities of the Wall Material Microparticles

Figure 3 shows the effect of microparticles prepared with wall materials only (no MAE) on the growth and AFB1 production by *A. flavus* NRLL 62477 strain.

The addition of increasing concentrations of wall material microparticles mildly modified the growth of *A. flavus*, likely related to the modification of available nutrients in the medium. Maltodextrin led to a slight reduction in growth (less than 4% at the highest tested concentration) whereas the two other wall materials mildly increased fungal growth. However, this had no significant impact on AFB1 production, with the exception of the highest concentration of soy protein microparticles (0.2 mg DM/mL), that slightly increased AFB1 production after 8 days of culture. This can be directly related to the increased fungal development observed at this concentration.

#### 2.4.2. Antifungal and Antiaflatoxin Activities of the Microparticles with MAE and Evolution after One Year pf Storage

The impact of MAE-containing microparticles on both fungal growth and AFB1 production at T0 and after one-year storage are presented on Figure 4.

The impact of MAE-containing microparticles on fungal growth at T0 was very limited (Figure 4A). Indeed, both starch and soy protein had no effect whatever the concentration tested. Only maltodextrin microparticles mildly reduced fungal growth (less than 5%) at all tested concentrations, as observed previously for particles without MAE. After one-year storage, quite comparable results were observed with no effect on fungal growth for both starch and maltodextrin at all tested concentrations. We were not able to test MAE-containing soy-protein microparticles after one year due to a lack of particles. Therefore, encapsulation of MAE in the different microparticles did not modify its limited impact on fungal growth.

Figure 4C shows the impact of MAE-containing microparticles on AFB1 production in *A. flavus*, compared to the effect of MAE at T0. Microparticles prepared with both polysaccharides had a similar inhibitory effect on AFB1 to MAE. Soy protein-based microcapsules were mildly less efficient in AFB1 inhibition. This could be the result of the balance between the stimulation of AFB1 production observed with soy protein alone (Figure 3B) and the effect of MAE active compounds on AFB1 synthesis.

After one year of storage, MAE-containing microparticles prepared with starch and maltodextrin were still able to inhibit AFB1 synthesis in a dose-dependent manner, with no significant reduction in their efficacy compared to what was observed at T0. By contrast, MAE extract stored for the same time at −20 °C lost part of its activity, since the maximal AFB1 inhibition was about 40% whereas it was higher than 65% at time 0.

In this study, we demonstrated that polysaccharide microparticles were able to preserve anti-AFB1 activity of MAE more efficiently than after storage at −20 °C. Since microparticles can be stored at room temperature with no special constraints, this may represent a good and energy-saving method of long-term storage of active components to be used on crops. The dry formulation is also of great interest considering grain protection during storage, meaning after a drying step. Indeed, in that case, active molecules from MAE would be released only in cases of moistening, which corresponds to an increase in the risk of fungal development and of subsequent toxin production.

## 3. Materials and Methods

### 3.1. Chemicals, Reagents and Encapsulating Agents

All solvents (HPLC grade) (acetonitrile; chloroform and methanol) and chemicals (Tween 80; Folin–Ciocalteu reagent; DPPH reagent; AFB1 standard; starch) were purchased from Sigma–Aldrich (Saint-Quentin-Fallavier, France). Ethanol 96% and sodium carbonate were purchased from VWR International (Fontenay-sous-bois, France). Ultrapure water was prepared using Veolia Purelab Classic (Veolia, Toulouse, France). Maltodextrin Glucidex IT12 was purchased from Roquette (Lestrem, France) and soy protein SUPRO^®^XT219DIP from Solae Belgium NV (Ieper, Belgium).

### 3.2. Plant Material

The supplier of stem bark of *M. tenuiflora* (Willd.) was Red Mexicana de Plantas Medicinales y Aromaticas (REDMEXPLAM) under the registration number UATX/01/Tepezcohuite/2019 deposited at Jardin Botanico Universitario UAT. At reception, bark was ground at 1 mm granulometry using a mill and then stored at 4 °C in sealed container.

### 3.3. Preparation of M. tenuiflora Aqueous Extract (MAE)

MAE was prepared as described [14]. *M. tenuiflora* bark was extracted by maceration for 15 h at room temperature. The resultant extract after centrifugation was filtered through Whatmann grade 1 filter paper (Sigma–Aldrich, Saint-Quentin-Fallavier, France). The sterilization was carried out at 121 °C for 20 min in an autoclave (SMI group UNICOM, Montpellier, France).

### 3.4. Dry Matter Content

Dry matter contents were determined for the extract and encapsulating agents by placing them at 103 °C until constant weight, using a Memmert oven (Schabach, Germany).

### 3.5. MAE Encapsulation by Spray-Drying

Infeed solutions were prepared as follows for all the encapsulating materials: the wall material was added to MAE under mechanical stirring (500 rpm) and the solution was heated at 70 °C for 30 min (still under stirring) to facilitate the solubilization of the wall material powder. The infeed solutions contained 0.65% (*w*/*w*) total solids, for a 50:50 core-to-wall ratio.

Freshly prepared solutions were then spray-dried using a Mini Spray-Dryer B-290 (Büchi, Flawil, Switzerland), in the open mode under the following stable conditions: inlet air temperature of 140 ± 4 °C, drying airflow rate of 470 L/h, liquid feed flow rate of 450 mL/h and 100% aspiration. Infeed solutions were maintained under magnetic stirring during the entire spray-drying step. Microparticles collected from the container were sealed in a hermetic opaque packaging and stored for analysis at room temperature. The spray-drying yield was calculated as follows:SP yield = [mass of microparticles recovered/Initial solid dry mass] × 100,(1)

### 3.6. Polyphenol Encapsulation Efficiency (PEE)

The encapsulation efficiency of the polyphenols was calculated as the experimental content of polyphenols contained in the microparticles in relation to the theoretical content of polyphenols.
%PEE = [Experimental content of polyphenols in microparticles/Theoretical content of polyphenols in microparticle] × 100(2)

The theoretical polyphenol content in the microparticles was calculated according to the polyphenol content of the extract and its proportional dilution in the wall material.

### 3.7. Scanning Electron Microscopy (SEM) Analysis

SEM observations of microparticles were performed using a Quanta 450 FEG scanning electron microscope (FEI, Hillsboro, OR, USA) under low vacuum mode at 100 Pa. The microparticles were deposited on conductive double-faced adhesive tape and sputter-coated with silver. They were observed at different magnifications (1000× and 8000×).

### 3.8. Solubilization of Microparticles for Analyses

The three wall materials are soluble in water, but not very soluble in ethanol. Therefore, the microparticles were first diluted in water, to carry out the tests on the fungus. For the determination of the antioxidant capacity, they were rediluted in ethanol as required by the technique, due to the solubility of the DPPH reagent in ethanol.

### 3.9. Determination of Total Polyphenol Content

The Folin–Ciocalteu method [33] was applied for the determination of total polyphenols content. The absorbance reading was performed at 700 nm after 45 min incubation time at 45 °C using a spectrophotometer BMG-Labtech SpectrostarNano (BMG LABTECH SARL, Champigny-sur-Marne, France). The results were expressed in milligrams of Gallic Acid Equivalent (GAE) per gram of dry matter.

### 3.10. Determination of Antioxidant Activity

DPPH (2,2-diphenyl-1-picrylhydrazyl) (Sigma-Aldrich) free radical was used to determine antioxidant activity of extracts and microparticles [34]. The reagent solution of DPPH in ethanol 96% was adjusted at a 516 nm absorbance of 1+/−10% in the same analyzing conditions as the extracts on BMG-LabtechSpectrostar-Nanospectrophotometer (BMG LABTECH SARL, Champigny-sur-Marne, France).

Seven different extract concentrations (150 µL) were incubated with 150 µL DPPH solution for 40 min at room temperature in a 96-well quartz microplate. The reaction equation for each reaction solution of five replicates was:*f*(*Ce*)*c* = 1 − *A*/*A*_0_(3)
where *Ce* is the extract concentration (in mg/L), *A* the extract absorbance, and *A*_0,_ is the control absorbance (DPPH) after 40 min incubation. The antioxidant activity is reported as the inhibitory concentration 50 (IC_50_), which corresponds to the concentration that inhibits 50% of the DPPH radicals. It is expressed in milligrams of extract per liter (mg/L). The value of IC50 was calculated using linear regression, as follows:IC_50_ = 0.5 − *a*/*b*(4)
where IC_50_ is the half-maximal inhibitory concentration, *a* is the origin ordinate, and *b* is the slope.

### 3.11. Effect of MAE and Microparticles on Aspergillus Flavus Growth and Aflatoxin B1 Synthesis

#### 3.11.1. Fungal Strain and Culture Conditions

The strain used for all assays was *Aspergillus flavus* NRRL 62477 [35]. Ten µL of a spore suspension containing 1000 spores were inoculated centrally into the culture medium for all experiments [14].

The culture medium was made of 18 mL of malt extract agar (Biokar Diagnostics, Allonne, France) and 2 mL of autoclaved MAE or microparticles extracts, prepared at different concentrations by dilution in water. Control cultures were made by adding 2 mL of distilled water to the initial 18 mL of culture medium. Cultures were incubated for 8 days at 27 °C. Each assay was performed in triplicate. After incubation, the growth was evaluated by the measurement of the colony diameter.

#### 3.11.2. Extraction and Quantification of AFB1 by HPLC

AFB1 was extracted from culture media with HPLC-grade chloroform and the supernatants were filtered off. Filtrates (2 mL) were concentrated to dryness at 45 °C using a STUART SBH200D/3 sample concentrator (Stuart equipment, Paris, France), and the residues were redissolved in acetonitrile HPLC grade (2 mL). Finally, the samples were filtered through 0.45 µm PTFE disk filters (Thermo Fisher Scientific, Illkirch-Graffenstaden, France) and placed in HPLC vials. AFB1 concentration was quantified by HPLC on an Ultimate 3000 UPLC equipment coupled to fluorescent detector at 365 nm wavelengths for excitation (and 430 nm for emission) (Thermo Fisher Scientific, Illkirch-Graffenstaden, France). Separation was carried out with an EvoC18 column (3 µm, 150 × 3.2, Phenomenex, Le Pecq, France) at 27 °C, at a 1.2 mL/min flow of a mixture acetonitrile HPLC grade and water (25:75 *v*:*v*). Injection volume was 10 µL.

The identity of the AFB1 was confirmed by checking the UV absorption spectrum thanks to an additional DAD detector in the series. AFB1 concentrations were calculated with an external standard calibration curve in the range of 0.16 to 20 mg/L.

### 3.12. Statistical Analysis

All data are presented as the mean of the experimental values obtained in quadruplicate. The error values of the data shown in tables as numerical values or in graphs as error bars correspond to the standard error of the mean (SEM) calculated from the standard deviation of the values.

One-way ANOVA was used to determine the differences in the AFB1 production and fungal growth data between the control and treated groups. A *T*-test was used to determine the significant difference between the means of two groups.

Differences were considered statistically significant for *p-*values less than 0.05.

GraphPad Prism 9 software was used to perform all statistical analyzes.

## 4. Conclusions

This work demonstrates that microencapsulation is a promising process to maintain the antioxidant and antiaflatoxigenic properties of natural extracts during long periods of storage. Among the wall materials tested here, polysaccharides appear to be the best choice since they are neutral on both fungal growth and toxinogenesis. Further studies are now required to test the applicability of such microparticles on grains during storage.

## Figures and Tables

**Figure 1 molecules-27-00496-f001:**
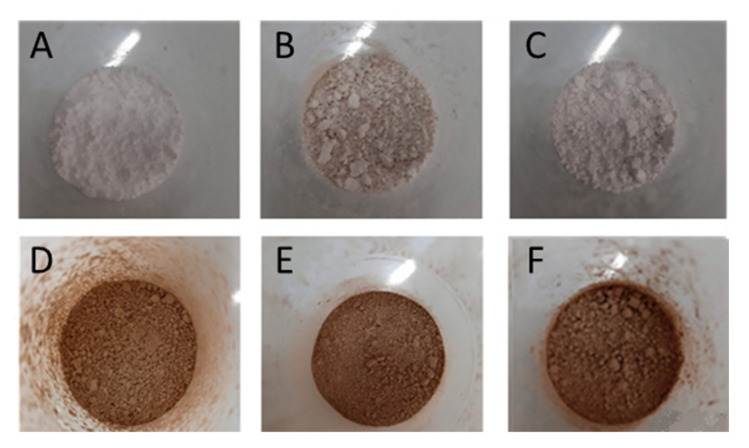
Visual aspect of the powders obtained after atomization in Mini Spray-dryer Buchi B-290 of wall materials alone: maltodextrin (**A**), soy protein (**B**), starch (**C**). Powders obtained after atomization of wall materials together with MAE are presented on the second line: maltodextrin + MAE (**D**), soy protein + MAE (**E**), and starch + MAE (**F**).

**Figure 2 molecules-27-00496-f002:**
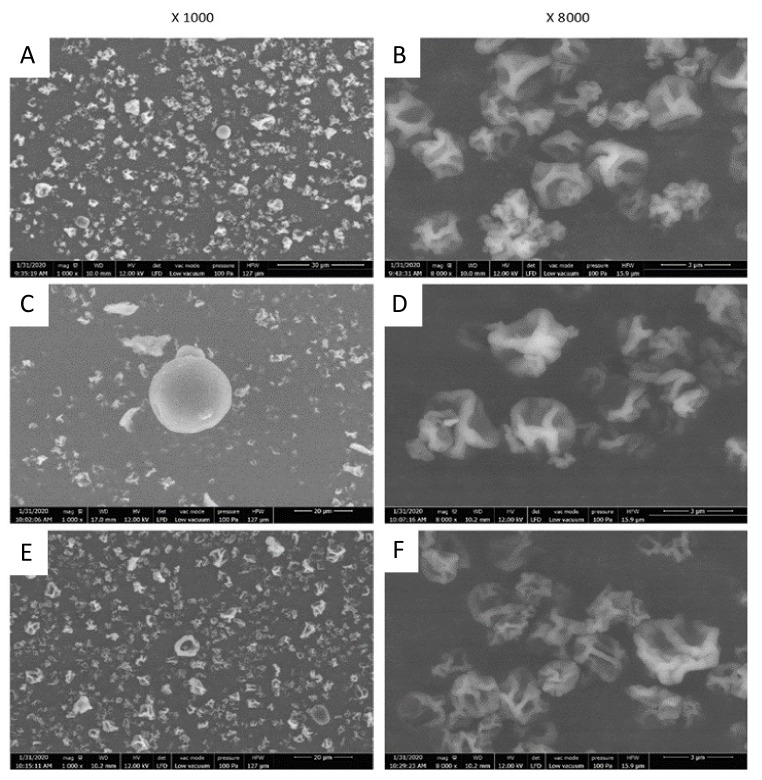
Microparticle analysis by SEM. Observations with a Scanning Electron Microscope of the microparticles obtained after MAE encapsulation in the three wall materials. (**A**,**B**) Maltodextrin microparticles with MAE, (**C**,**D**) soy protein microparticles with MAE, (**E**,**F**) starch microparticles with MAE.

**Figure 3 molecules-27-00496-f003:**
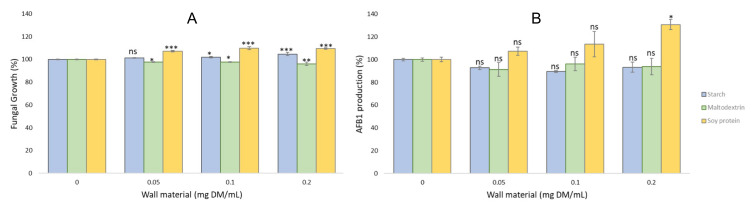
Fungal growth (**A**) and AFB1 production (**B**) of *A. flavus* NRRL 62477 strain when exposed to increasing concentrations of the different wall-material microparticles for 8 days at 25 °C: Starch (blue bars), maltodextrin (green bars) and soy protein (yellow bars). Columns with 0 correspond to *A. flavus* NRRL 62477 cultures with no wall materials and are used as control values. Results are expressed as the percentage of fungal growth or AFB1 production compared to untreated control cultures ± standard error of the mean (*n* = 4). ns = no significant change; * *p-*value < 0.1; ** *p-*value < 0.01; *** *p-*value < 0.001.

**Figure 4 molecules-27-00496-f004:**
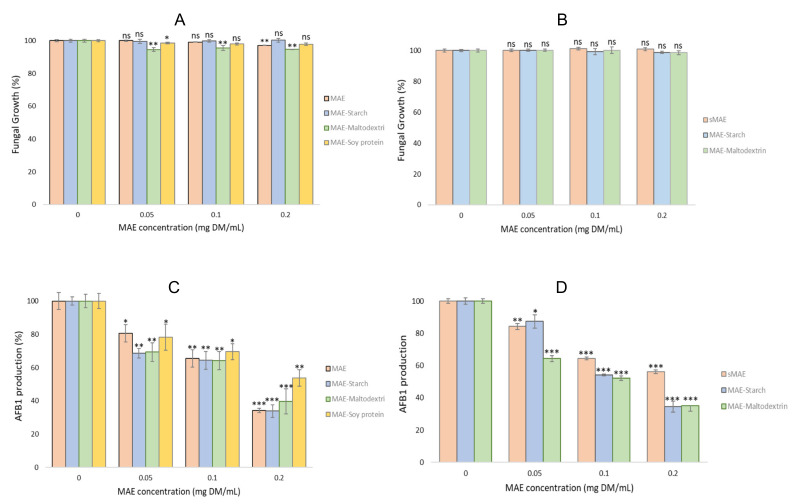
Fungal growth at time 0 (**A**) and after one-year storage (**B**). AFB1 production at time 0 (**C**) and after one-year storage (**D**). *A. flavus* NRRL 62477 strain was exposed to increasing concentrations of MAE-containing microparticles for 8 days at 27 °C. MAE (**A**,**C**) and sMAE (**B**,**D**) were used as a positive control at T0 and after one year of storage, respectively. Columns with 0 correspond to *A. flavus* NRRL 62477 cultures with no particles nor MAE or same, and are used as control values. Results are expressed as the percentage of fungal growth or AFB1 production compared to these untreated control cultures ± standard error of the mean (*n* = 4). ns = no significant change; * *p-*value < 0.1; ** *p-*value < 0.01; *** *p-*value < 0.001.

**Table 1 molecules-27-00496-t001:** Characterization of the aqueous extract at time 0 and after one year of storage at −20 °C.

	Dry Mass (%)	Polyphenol Content (mg GAE/g DM Extract)	Antioxidant Activity ^b^ IC50 (mg/L)	Maximum AFB1 Inhibition ^c^ (%)	Maximum Fungal Inhibition ^c^ (%)
MAE	11.2	397 ± 22	10	65.7	3
sMAE ^a^	11.1	376 ± 2 ^ns^	12	43.8 *	1 ^ns^

^ns^ no statistically significant change; * statistically significant change: *p-*value < 0.05. ^a^: sMAE: MAE stored one year at −20 °C. ^b^: measured using DDPH assay. ^c^ measured after incubation of *Aspergillus flavus* NRRL 62477 with 0.2 mg DM/mL for 8 days at 27 °C.

**Table 2 molecules-27-00496-t002:** Spray-drying yields for the different wall materials tested.

Composition of the Microparticles	Wall Material/Extract Ratio (*w*/*w* Dry Matter)	Spray-Drying Yield (%)
Maltodextrin	100/0	59
Maltodextrin + MAE	50/50	63
Soy protein	100/0	57
Soy protein + MAE	50/50	54
Starch	100/0	58
Starch + MAE	50/50	59

**Table 3 molecules-27-00496-t003:** Total polyphenol content and antioxidant activity of wall material microparticles.

Wall Material	Polyphenol Content (mg GAE/g DM)	Antioxidant Activity on DPPH IC50 (mg DM/L)
Maltodextrin	0.7 ± 0.3	>1000
Soy protein	17.0 ± 0.4	>1000
Starch	1.9 ± 0.4	>1000

**Table 4 molecules-27-00496-t004:** Polyphenol content and antioxidant activity of the microparticles with MAE and evolution after one-year storage at room temperature.

Wall Material Used	Theoretical Dry Mass of the Extract in the Microparticles (g)	Theoretical Polyphenol Content in Microparticles (mg GAE/g Microparticles)	Time 0			After One-Year Storage ^c^	
			Experimental Polyphenol Content in Microparticles (mg GAE/g Microparticles)	PEE ^a^ (%)	Antioxidant Activity IC50 ^b^ (mg/L)	Experimental Polyphenol Content in Microparticles (mg GAE/g Microparticles)	Antioxidant Activity IC50 ^b^ (mg/L)
Maltodextrin	0.65	258	168 ± 7	65	23	188 ± 3	27
Starch	0.62	246	214 ± 8	86	25	211 ± 3	27
Soy protein	0.62	246	142 ± 9	58	48	149 ± 3	79

^a^: PEE = Polyphenols encapsulation efficiency. ^b^: measured using DDPH assay. ^c^: storage at room temperature.

## Data Availability

The data presented in this study are available on request to the corresponding author.

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
