# Peer review of "Preservation of Mimosa tenuiflora Antiaflatoxigenic Activity Using Microencapsulation by Spray-Drying"

_molecules, 2022, doi:10.3390/molecules27020496_

Round 1
Reviewer 1 Report
Congratulations. Your work have a very good potential of transfer in food and feed industry , with very beneficial results for human health. An EU grant project on this subject, involving both research and industry parteners is recommended.
Author Response
the authors thank the reviewer very much for this comment
Reviewer 2 Report
The paper entitled ''Preservation of Mimosa tenuiflora anti-aflatoxigenic activity using microencapsulation by Spray Drying'' by Hermandez et all sound very scientific and presents nice piece of work. Despite its relatively good quality of the work presented with this study, I would kindly ask authors to elaborate more on the way how the error bars within the presented graphs were exactly calculated. The software they used for this purpose is mentioned, but we all know how the statistical analysis is important in our research work that is why I think that more details should be presented with respect to these calculations.
Moreover, within introduction authors should browse more the literature (especially the new one) about the new reports on the natural products and their reported activities. I would suggest to look at the following papers and refer to them:
Sci Rep. 2021 Jul 15;11(1):14539. doi: 10.1038/s41598-021-93285-7.
Antibiotics (Basel). 2021 Jul 21;10(8):887. doi: 10.3390/antibiotics10080887.
Molecules. 2021 Mar 8;26(5):1459. doi: 10.3390/molecules26051459.
English should be a little bit improved.
Author Response
Comment: The paper entitled ''Preservation of Mimosa tenuiflora anti-aflatoxigenic activity using microencapsulation by Spray Drying'' by Hermandez et all sound very scientific and presents nice piece of work.
Answer: The authors thank the reviewer for his appreciation of our work.
Comment: Despite its relatively good quality of the work presented with this study, I would kindly ask authors to elaborate more on the way how the error bars within the presented graphs were exactly calculated. The software they used for this purpose is mentioned, but we all know how the statistical analysis is important in our research work that is why I think that more details should be presented with respect to these calculations.
Answer: Additional information on the error bars determination was added in the revised version (paragraph 3.12, lines 441-449 of the revised version)
Comment: Moreover, within introduction authors should browse more the literature (especially the new one) about the new reports on the natural products and their reported activities. I would suggest to look at the following papers and refer to them:
Sci Rep. 2021 Jul 15;11(1):14539. doi: 10.1038/s41598-021-93285-7.
Antibiotics (Basel). 2021 Jul 21;10(8):887. doi: 10.3390/antibiotics10080887.
Molecules. 2021 Mar 8;26(5):1459. doi: 10.3390/molecules26051459.
English should be a little bit improved.
Answer: The mentioned papers are very interesting but quite far from the research area of our work. Indeed, the two first ones deal with antileishmanial activity and antibacterial activity of metabolites from Streptomyces smyrnaeus (no plant product, no fungi, no mycotoxin) and the last one is about the characterization of polysaccharides from one fungus but has no link with the use of such compounds as encapsulating agent.
Nevertheless, since it appears that indeed, the introduction could be improved, we added 8 references (number 9, 10, 12, 13, 15, 16, 19, 20 in the revised version) among which 4 published in 2021 in order to make the introduction section clearer and more accurate.
Some sentences were also rephrased to make them more understandable.
Reviewer 3 Report
The authors evaluate the preservation of Mimosa tenuiflora anti-aflatoxigenic activity using microencapsulation by Spray Drying. The study is promising and well informative. However, I do not recommend the manuscript for publication in this present form. the manuscript can be improved. There are problems with careless writing (misspellings, typographical errors, and faulty punctuation) in the manuscript as well as omission of some important information and in appropriate citation by the author in the method section. I suggest that authors should carefully edit the paper. Please see specific comments below.
Title
The title of the study is fine, it fits perfectly with the description of the study.
Abstract
The abstract is well structured and articulated.
Introduction
The introduction is fairly expressed. It can be improved. What has been done previously on the investigated plant was not well stated. Authors need to include what has been done previously on the plant (Mimosa tenuiflora). Also, there are some sentences in the introduction that need better clarity. For examples, the sentence in line 52 to line 53 needs to be rephrased for better understanding.
Line 61: information omitted should be written in full. Author should complete it.
Line 76: it should be “maltodextrin” not maldotextrin.
Result and discussion
The title of figure 1 need to be rephrased for convenient reading. In figure 3 and 4, Author should include the untreated control bar (untreated control which should be at 100%) for better comparison between the tested samples and the untreated control.
Material and methods
The method is structured and reproducible. However, some sentences are not clear. For example, sentence in line 372 and 373 should be rephrased. Also, there is in appropriate citation by the author in the method section. Authors should try to cite another author instead of Hernandez et al., 2021 in the method section.
Author Response
The authors evaluate the preservation of Mimosa tenuiflora anti-aflatoxigenic activity using microencapsulation by Spray Drying. The study is promising and well informative. However, I do not recommend the manuscript for publication in this present form. the manuscript can be improved. There are problems with careless writing (misspellings, typographical errors, and faulty punctuation) in the manuscript as well as omission of some important information and in appropriate citation by the author in the method section. I suggest that authors should carefully edit the paper. Please see specific comments below.
Title
The title of the study is fine, it fits perfectly with the description of the study.
Answer: the authors thank the reviewer for this comment
Abstract
The abstract is well structured and articulated.
Answer: the authors thank the reviewer for this comment
Introduction
The introduction is fairly expressed. It can be improved. What has been done previously on the investigated plant was not well stated. Authors need to include what has been done previously on the plant (Mimosa tenuiflora). Also, there are some sentences in the introduction that need better clarity. For examples, the sentence in line 52 to line 53 needs to be rephrased for better understanding.
Answer: the authors improved the introduction section as recommended. Some recent references (6) were included, the nature of the effect of Mimosa tenuiflora extract was better described and the sentence that was previously on lines 52-53 was rephrased to make it more understandable (see lines 53-61 of the revised version)
Line 61: information omitted should be written in full. Author should complete it.
Answer: sentence was rephrased to make it clearer
Line 76: it should be “maltodextrin” not maldotextrin.
Answer: text was corrected accordingly, sorry for the typing error
Result and discussion
The title of figure 1 need to be rephrased for convenient reading.
Answer: the legend of the figure 1 was rephrased accordingly to be more readable.
In figure 3 and 4, Author should include the untreated control bar (untreated control which should be at 100%) for better comparison between the tested samples and the untreated control.
Answer: as recommended by the reviewer, we added in both figures the results for control cultures (in the absence of wall material, microparticle or MAE).
Material and methods
The method is structured and reproducible. However, some sentences are not clear. For example, sentence in line 372 and 373 should be rephrased. Also, there is in appropriate citation by the author in the method section. Authors should try to cite another author instead of Hernandez et al., 2021 in the method section.
Answer: as recommended by the reviewer, we rephrased the sentence (see line 427-427 of the revised version) and the inappropriate reference was removed from the text.
Round 2
Reviewer 3 Report
1) In figure 3A and B, there is no control bar/column. Authors should include it.
2) Figure 3B is not clear and visible for the reader. Authors should delete the figure 3B and include clear figure of 3B.
3) There are too many figures in figure 4. There is absolutely no need for duplicate. I will suggest that authors should delete figure 4ABC and D without control bar/column (0) and leave the figures 4ABC and D with control bar/column (0). Those figures (4ABC and D with control bar/column) should be very clear and visible for readers.
4) Line 336 sentence should read as MAE was prepared as described [14]
Author Response
The authors would like to apologize since it seems that the "track changes" version of the revised version led to some troubles in reading/seing modifications that were done. It shall probably explain why controls were not visible on figure 3 whereas they had been included and also why figure 4 was included twice (former and revised version of the figure).
We resubmitted a "clean" version of the paper with only the correct figures that correspond to previous remarks/demands of the revewier (meaning with control bars included).
We also corrected the sentence line 345 of the revised version as recommended.